# Mapping the glycosyltransferase fold landscape using interpretable deep learning

Rahil Taujale[1,2,5], Zhongliang Zhou [3,5], Wayland Yeung[1], Kelley W. Moremen [2,4], Sheng Li[3] & Natarajan Kannan [1,4✉]

Glycosyltransferases (GTs) play fundamental roles in nearly all cellular processes through the biosynthesis of complex carbohydrates and glycosylation of diverse protein and small molecule substrates. The extensive structural and functional diversification of GTs presents a major challenge in mapping the relationships connecting sequence, structure, fold and function using traditional bioinformatics approaches. Here, we present a convolutional neural network with attention (CNN-attention) based deep learning model that leverages simple secondary structure representations generated from primary sequences to provide GT fold prediction with high accuracy. The model learns distinguishing secondary structure features free of primary sequence alignment constraints and is highly interpretable. It delineates sequence and structural features characteristic of individual fold types, while classifying them into distinct clusters that group evolutionarily divergent families based on shared secondary structural features. We further extend our model to classify GT families of unknown folds and variants of known folds. By identifying families that are likely to adopt novel folds such as GT91, GT96 and GT97, our studies expand the GT fold landscape and prioritize targets for future structural studies.

[1] Institute of Bioinformatics, University of Georgia, Athens, GA, USA. [2] Complex Carbohydrate Research Center, University of Georgia, Athens, GA, USA. [3] Department of Computer Science, University of Georgia, Athens, GA, USA. [4] Biochemistry and Molecular Biology, University of Georgia, Athens, GA, USA. [5] These authors contributed equally: Rahil Taujale, Zhongliang Zhou. ✉email: nkannan@uga.edu

Glycosyltransferases (GTs) are a large family of enzymes tasked with the biosynthesis of complex carbohydrates that make up the bulk of biomass in cells[1]. Prevalent across the tree of life, these enzymes catalyze the transfer of a sugar molecule from an activated donor (mostly nucleotide sugars or dolichol-(pyro)phosphate linked sugars) to a wide variety of acceptors ranging from proteins and fatty acids to other carbohydrate molecules. The CAZy database[2] classifies over half a million GT sequences across organisms into 114 families based on overall sequence similarity. While sequences within families share detectable sequence similarity, sequences across families share little or no similarity[3]. The extensive diversification of GT sequences presents a major bottleneck in investigating the relationships connecting sequence, structure, fold, and function.

As with other large protein families, GTs also exhibit much higher conservation in 3D structural fold compared to primary sequences[4–6]. Across all 114 families, only 3 major folds have been identified (GT-A, -B, and -C folds) with some families adopting other unique folds[1,7,8]. Currently, around 34 GT families are classified as GT-A fold, 32 families as GT-B and 10 families as GT-C. In addition, a large family of peptidoglycan polymerases of the GT51 family has been known to adopt a unique lysozyme-type fold[9]. Recently, we proposed a phylogenetic framework relating diverse GT-A fold enzymes leveraging the common structural features identified through structure-guided curation of large multiple sequence alignments[3]. While such multiple sequence alignment-based approaches have provided insights into GT-A fold structure and evolution, such approaches are not scalable to other GT folds for which there is limited structural data or limited structural homology.

The recent explosion of deep-learning methods, in particular multilayer neural networks, offer new opportunities for sequence classification and fold prediction through feature extraction and pattern recognition in large complex datasets[10,11]. The most recent successful application of these methods has been in the area of protein structure prediction in which the deep-learning model extracts residue co-variation from multiple sequence alignments to predict residue contacts in 3D structures[12–18]. Of note is Alphafold2[19], an attention-based model that significantly outperformed other structure prediction methods in the biennial CASP assessment[20]. Other related efforts have focused on making residue level predictions such as disorder, solvent accessibility, and post-translational modifications[21–24] using evolutionary information encoded in multiple sequence alignments. In these applications, the accuracy of predictions relies heavily on the quality of input multiple sequence alignments and these models cannot be directly extended for the study of divergent protein families such as GTs for which generating accurate multiple sequence alignments is a challenge for reasons mentioned above. Furthermore, the black-box nature of existing deep-learning models prevents a direct biological interpretation of sequence or evolutionary features contributing to structure or fold prediction.

Here, we report a convolutional neural network[25] with attention (CNN-attention)-based model for GT-fold type prediction solely based on secondary structure annotations as input. These coarse-grained input features are based on the premise that protein secondary and tertiary structures are far more conserved than primary sequences. Our model makes no use of amino acid physicochemical properties nor does it rely on generating evolutionary or alignment-based information and yet, achieves an average accuracy of 96% on fold prediction, and 77% on family classification. In contrast, other methods such as the Hidden Markov model (HMM)[26], Long Short-term Memory (LSTM)[15], and other CNN-based methods[16] had a much lower accuracy for both fold and family classification. By using specially designed attention[27] modules, the trained model can generate highly interpretable activation maps that help locate conserved segments within sequences that point to the common cores within folds. We further leverage recent advances in open set recognition[28] and use a specially modified reconstruction error loss term to determine similarities between GTs so as to expand our model beyond known GT folds. The major advantages of our model are threefold: (1) We propose an alignment-free method to explore protein folds by leveraging secondary structure prediction as input data. (2) We focus on the interpretability of the model to mine features learned by the model and make meaningful biological inferences. (3) We extend our trained model to make predictions on GT families with unknown folds and report the ones most likely to adopt novel fold types to guide further research on the discovery of novel glycoenzymes. The approach is applicable to other broad, heterogeneous protein families where challenges in primary sequence alignment approaches have hindered the analysis of fold classification and evolutionary relationships.

## Results

### A deep-learning framework to identify, classify, and predict glycosyltransferase folds.

We first sought to develop a deep-learning model that could distinguish the features of glycosyltransferase (GT) structural folds from a large amount of readily available sequence information. To this end, we collected over half a million GT sequences from the CAZy database and filtered them based on sequence similarity, length, and other criteria (see "Methods") to generate a representative set of 44,620 GT sequences spanning all folds and families for training. Previous large-scale analysis of GTs has revealed that the overall organization of the secondary structures is far more conserved within folds than primary sequences[3,29]. Therefore, we identified secondary structure patterns using NetSurfP2.0[24] and used them as the only input to train a six-layer CNN model for multitasking fold and family classification (Fig. 1 and Supplementary Fig. 1). After refinement by the addition of attention modules and data augmentation strategies ("Methods"), the final optimized model achieves fold prediction with 96% accuracy and family classification with 77% accuracy, based on tenfold cross-validation. Results for this final model highlighting the effects of data augmentation and the addition of multitasking and attention modules are provided in Supplementary Table 1. We also compared our model with several other alternative methods that are routinely applied in protein classification problems such as secondary structure-based HMM searches[26] (Supplementary Table 2), a Long short-term memory (LSTM) model[15], another CNN-based model[16], and a more recent transformer-based embeddings model with GDBT classifier[17,18] (Supplementary Table 3). These comparisons further illustrate the efficiency of our CNN-attention model both in terms of accuracy and interpretability. While the transformer model achieves comparable accuracy, its generated projections do neither separate the GT folds as efficiently (Supplementary Fig. 2) nor is the model interpretable.

The first three layers of our CNN model (Block 1, Fig. 1) learn different levels of patterns in conserved secondary structure features guided by the class labels. These features are stored as layer-specific weights along with their spatial resolution enabling the use of methods such as Class-specific Activation Maps using Grad-CAM[30,31] (CAM) to project them back into the linear sequences and 3D structures. This projection assigns CAM values to specific residues within sequences and structures where high CAM values correspond to residue positions that distinguish them the most from other class labels (folds and families). Thus, CAM values can be used to identify the distinguishing features of a given GT fold recognized by the model. The last three layers

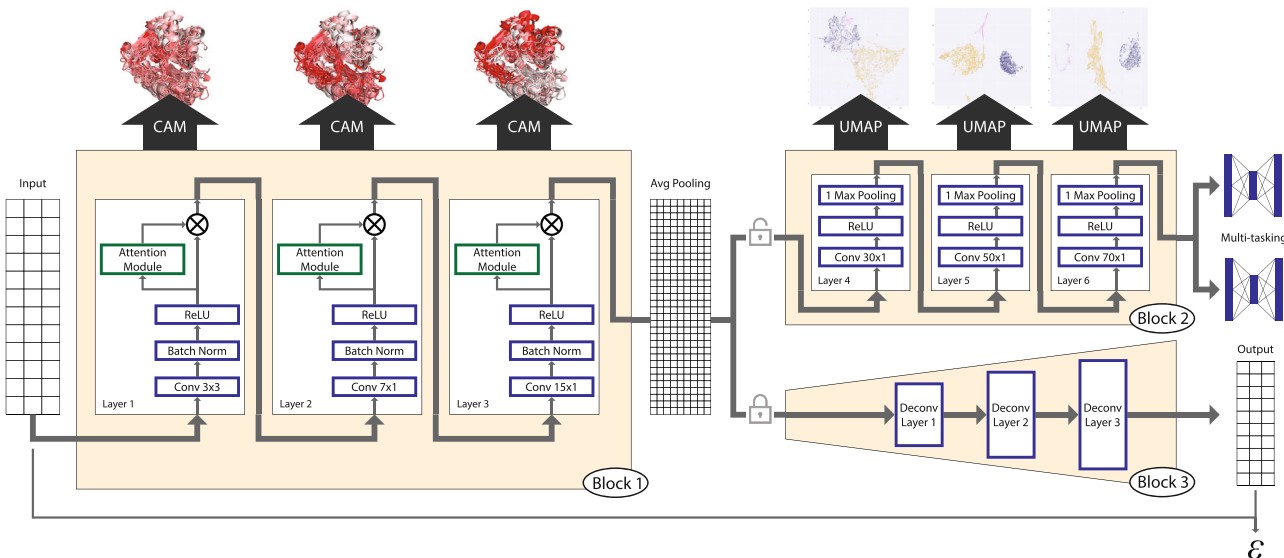

**Fig. 1 Overall schematics of the deep-learning model used.** A three-state secondary structure prediction matrix for each sequence is used as input to Block 1. Block 1 includes the first three sequential one-dimensional convolutional layers with attention for feature maps refinement. Feature maps from Block 1 are passed through a global average pooling for dimension reduction and fed into Block 2 with three additional convolutional layers, and finally used to make predictions for both fold and family. Blocks 1 and 2 constitute the deep CNN model for classification. Using Grad-CAM, features from Block 1 are mapped back into sequences and structures for interpretation. Features from Block 2 are passed to UMAP for dimensionality reduction and visualization. Weights and features from Block 1 are frozen and used in an encoder that is passed to Block 3 which is the decoder with multiple deconvolution steps that complete an autoencoder model. Reconstruction error ($\varepsilon$) from this model is used to make predictions of fold type on GT families with unknown folds (GT-u).

(Block 2, Fig. 1) further optimize the associated feature weights before feeding them into a fully connected multitask classifier to generate a classification with high accuracy. We extract these optimized feature embeddings and analyze them using dimensionality reduction by Uniform Manifold Approximation and Projection (UMAP)[32] to visualize the classification (Supplementary Fig. 3). In contrast to the more prevalent black-box deep-learning models, this architecture results in a highly interpretable model[33] with quantitative outputs to evaluate each step with high scrutiny and draw meaningful insights into secondary structure patterns associated with GT function and fold.

These two blocks also enable us to classify GT families of unknown structures into known folds or assign them to novel folds. To classify GT families of unknown structure or fold, we integrate an autoencoder framework to our existing model in which the optimized weights from Block 1 are frozen and used as a general feature extractor for the encoder. Block 3 (Fig. 1) is then designed as a decoder with a mirror structure of the CNN model that performs deconvolution operations. Applying the concepts of open set recognition framework that aim to extend knowledge from observed samples (closed set) to unseen samples (open set), we generate reconstruction errors (RE) using a modified mean square error, which measures how close a sequence with an unknown fold is to one of the known GT folds used in training ("Methods")[34]. This measure is then used to identify GT families that are most likely to adopt novel folds. We discuss the results from the three blocks of our model in the following sections.

**A landscape of all GT folds reveals distinct clusters within major fold types.** We visualized feature maps generated from the three layers of Block 2 with the UMAP algorithm[32] (Fig. 2a). As expected, we find separations between all the major GT folds, highlighting the model's ability to distinguish them. Sequences from the same GT family cluster together throughout, indicating the conservation of secondary structures and the overall fold within individual families (Supplementary Fig. 4, bottom panel).

Moreover, we find distinct substructures for the GT-A, -B, and -C fold types. To further analyze these substructures, we first ran separate UMAP analyses on each of the threefold types and clustered the resulting projections using the Gaussian Mixture Model (GMM) algorithm[35] to identify clusters within the major GT fold types. This resulted in two GT-A clusters and three GT-B and GT-C clusters.

The two distinct GT-A clusters accounted for most of the families with 17 out of 34 families grouping into a larger GT-A0 cluster. Ten families were grouped into the GT-A1 cluster, while the remaining seven families did not group and scattered away from the two central clusters (Fig. 2b). Sixteen out of 32 GT-B families used in training fall within the central GT-B0 cluster, while other families are spread out into smaller subclusters (five families in GT-B1, six in GT-B2, and five families ungrouped) (Fig. 2c). Likewise, GT-C sequences are also scattered across three major clusters (Fig. 2d) with only two out of ten families (Alg10 glucosyltransferases of GT59 and the bacterial GT85 family) not grouped into any of the three clusters. We discuss the structural basis for the separation of these GT-A, -B, and -C clusters in the following sections. In contrast, the lysozyme-type GT fold sequences (GT-lyso) all cluster into a single compact cluster, indicating the structural similarity within this fold type and its stark difference from all the other fold types. A list of families belonging to each of these identified clusters is provided in Supplementary Table 4 and their placement in the clusters within the UMAP projections are labeled in Supplementary Fig. 5. The 2D UMAP projection also shows several outlier sequences that do not fit within individual clusters. These sequences were either fragments that lack an entire GT domain or display secondary structure patterns significantly different from related family members (Supplementary Fig. 6).

**CAM maps for GT-A clusters highlight differences in shared structural features.** In order to understand the structural features of the major GT folds and their respective clusters, we mapped

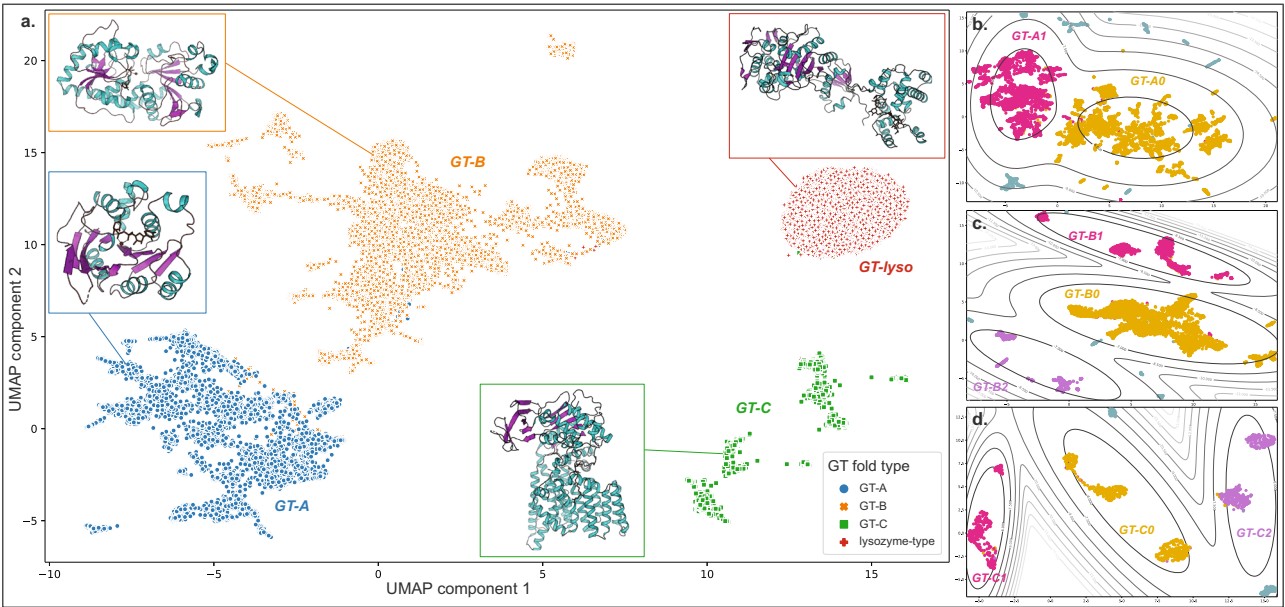

**Fig. 2 UMAP projection shows the separation of the major GT fold types.** Dots represent 2D UMAP projection of features for individual sequences. UMAP plot was generated across a scan of parameters (n_neighbors = 5, 15, 20 and min_dist = 0.001, 0.01, 0.1, 0.5) with three plots for each combination of parameters to ensure reproducibility (Supplementary Fig. 3). **a** Dots are colored based on their fold type and show a clear separation. Representative structures for each fold type are also shown. UMAP was applied separately on each major fold type and the projections for sequences belonging to the GT-A (**b**), GT-B (**c**), and GT-C (**d**) folds are shown. Clustering was done on these projections based on a Gaussian Mixture Model (GMM). Gray lines represent the contour for GMM scores around each cluster. Sequences that belong to a cluster are colored in yellow, magenta, or purple and are labeled with the cluster name. Sequences that do not belong to any cluster are colored in teal. Source data are provided as a Source Data file.

the CAM values obtained from each of the first three layers of the CNN model back to their respective sequences. In our previous work[3], we identified a common core shared by all GT-A fold enzymes. We first mapped the CAM values back to this GT-A common core alignment (Fig. 3a, b). We find that the regions with the highest conservation in the GT-A core (such as the DXD motif, G-loop, and the first two beta-sheets of the characteristic Rossmann fold) correspond to the regions with the highest CAM values, indicating that the model is using these conserved regions to distinguish the GT-A fold from other GT fold types. It is important to emphasize that while our previous analysis required a laborious curation of the profiles and alignment to identify these regions, our current CNN-attention model was able to recognize and utilize these regions without any prior information or sequence alignment but only based on the predicted patterns of conserved secondary structures across sequences.

CAM maps generated from layer 2 were the most informative and matched well with the core features of the GT-A fold. Layer 1 CAM values correspond to minute regions scattered throughout the domain and likely indicate local features learned by the model while CAM values from layer 3 extend over longer contiguous regions (Fig. 3c), possibly capturing long-range correlations.

UMAP projection and clustering indicate the presence of two GT-A clusters (Fig. 2b). GT-A cluster 0 (GT-A0) primarily constitutes large and phylogenetically distinct GT-A families such as GT2 and GT8 along with their closely related counterparts like GT84 (β-1,2-glucan synthases), GT21 (ceramide β-glucosyltransferases), and GT24 (glycoprotein α-glucosyltransferases) (Supplementary Table 4). This cluster includes more than half of all the GT-A sequences used in training and represents a consensus secondary structure that most closely matches the conserved core of the GT-A fold. The GT-A1 cluster includes GT31 and closely related families like GT15 and GT67. It also includes phylogenetically and functionally diverse families like GT7, GT77, and GT6. Meanwhile, families such as GT88 (bacterial Lgt1 sequences known to include large multi-helix

insertions[36]), GT75 (that includes the self-glucosylating β-glucosyltransferases and UDP-L-arabinopyranose mutases), GT54 (MGAT4), and a few others are isolated away from the two main clusters (Supplementary Fig. 5), indicating some distinction in their secondary structure patterns from other GT-A families.

In contrast to the GT-A1 consensus, GT-A0 families are distinguished by helical segments: the first one in the hypervariable region 2 (HV2) preceding the G-loop and the second one in the C-terminal HV3 region following the C-His position (Fig. 3c). Both of these helices have been previously shown to harbor family-specific residues directly involved in donor or acceptor binding[3]. The ability of our model to cluster the evolutionarily divergent GT-A0 families based on the conservation of these helices highlights the value of our CNN-attention model in identifying convergent substrate-binding mechanisms that are difficult to infer using traditional phylogenetic approaches.

**The multiple levels of conserved core in GT-B and GT-C clusters.** Our analysis identified a large central GT-B cluster (GT-B0) that includes some of the largest GT families such as GT4 with diverse functions and donor substrates, the UDP glucose/glucuronosyltransferases of GT1, GT5 sequences involved in glycogen and starch biosynthesis, and lipopolysaccharide GlcNAc transferases of the GT9 family. Other families that cluster together include the fucosyltransferases from GT10 and GT37, trehalose phosphate synthases from GT20, the xylosyl-/glucosyltransferases from GT90 and others. Clearly, families with a variety of functions including the largest and one of the most ancient families (GT4, which is also present in Archaea) are grouped together into a single cluster suggesting shared structural similarities within the GT-B fold. We additionally identify two other GT-B clusters, GT-B1 and GT-B2, both of which are slightly sparser than GT-B0 and include fewer families (Supplementary Fig. 5). To further expand on the structural similarities

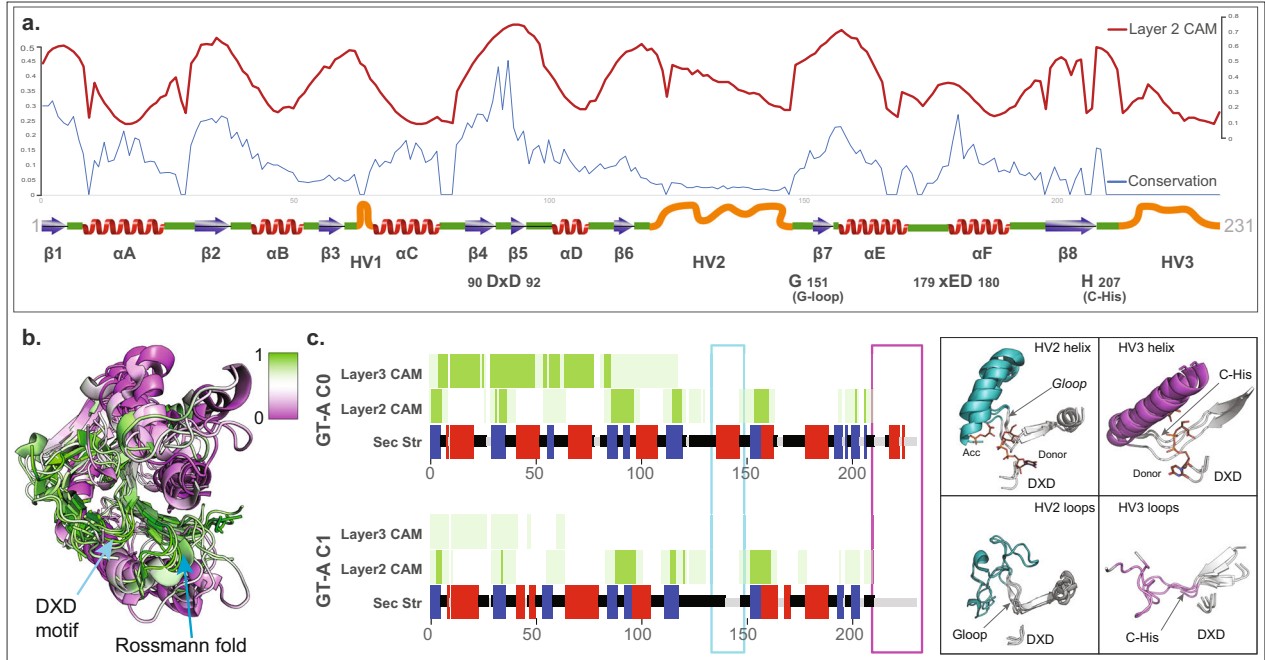

**Fig. 3 CAM highlights the GT-A fold core. a** The activation values from layer 2 are plotted in red line on top of the conserved secondary structures (blue arrows: beta-sheets; red: helices; green: loops, orange: hypervariable regions) and the conservation scores in blue line. The conservation scores and GT conserved core schematics were collected from ref. [3]. The most conserved regions generally have higher activation values. **b** CAM values are mapped onto a structural alignment of the GT-A conserved core. The conserved regions are shown to have a high CAM value indicated by high intensity of green and low CAM value by purple. **c** Left: consensus secondary structure for the aligned positions in the two GT-A fold clusters are shown (blue: beta-sheets; red: helices; green: loops). Average CAM values from layer 2 and layer 3 of the CNN-attention model are shown for each aligned position (higher intensity of green corresponds to a higher CAM value). Cyan and magenta boxes highlight the secondary structure differences between the two clusters near the HV2 and HV3 region respectively. Right: the regions with differences in secondary structure are shown in representative structures from each cluster (GT-A0: GT81 family structures 3ckq, 3o3p, 4y6n; GT-A1: GT6 family structures 5c4b, 5nrb, and GT7 family structures 2ae7, 4lw6) and highlighted in cyan and magenta. The conserved DXD motif, G-loop and C-His are indicated for reference. Donor and acceptor substrates for GT-A0 are shown as sticks. Source data are provided as a Source Data file.

shared within members of these clusters, we compare the CAM maps obtained for each of the GT-B fold families.

While it has been especially challenging to generate a GT-B fold-wide sequence alignment due to the lack of sequence conservation, in order to understand the patterns obtained from our CNN model, we generated family-level alignments for each of the GT-B families. We then calculated a consensus secondary structure and average layer 2 CAM map (Fig. 4a) for each family. All of these families reflect the typical two β/α/β Rossmann-fold domains characteristic of the GT-B fold. The most consistent pattern picked up by the CNN-attention model is the C-terminal Rossmann fold. Features associated with its 6 beta-sheets are always significant in distinguishing GT-B families as indicated by the CAM value maps (cyan box in Fig. 4a) and the conservation of this C-terminal region extends beyond GT-B0 to GT-B1, GT-B2, and other ungrouped GT-B families as well. Further, mapping the CAM values to representative structures revealed that the C-terminal Rossmann-fold orientation and structure is well-conserved across GT-B families with occasional family-specific insertions in the loop regions (Fig. 4b). Thus, our study supports the C-terminal Rossmann domain as the common structural feature of GT-B fold families.

Upstream of the C-terminal Rossmann fold, CAM values are also higher in the secondary structure of the N-terminal Rossmann-fold region, likely indicating its importance in distinguishing the GT-B fold with 2 Rossmann folds versus the GT-A fold that has only a single Rossmann-fold domain. However, these CAM value patterns are not consistent across families. Most families have a different number and order of beta-sheets, suggesting variability in the N-terminal domain, likely reflecting its function of binding different

types of acceptor substrates, as shown in the previous studies[1,37]. This variability is especially prominent in GT-B1 where families accommodate additional secondary structures in the N-terminal (e.g., tetratricopeptide repeats in GT41 and coiled coils in GT23) (Fig. 4a Supplementary Fig. 7). Conversely, all families within the GT-B2 cluster are found to conserve a minimum of six beta-sheets and five alpha-helices in the N-terminal Rossmann fold, as indicated by the CAM values (magenta box in Fig. 4a, b), highlighting the extension of the GT-B2 core to include both the N- and the C-terminal Rossmann fold domains. The functional implications of this extended core conservation in GT-B2 families is yet to be determined.

GT-Cs present an entirely different fold composed of 8–13 hydrophobic integral transmembrane helices with the active site and catalytic residues in long loop regions that makes them stand out from other GT fold enzymes[29]. The layer 3 CAM values of our CNN model responsible for capturing long-range features recognized this trend and presented a consistent pattern for distinguishing the GT-C fold families (Fig. 4c). In contrast to GT-A and GT-B, no other trends in CAM values from layers 1 and 2 exist for the GT-Cs, suggesting that the layer 3 features were the most important and sufficient in distinguishing sequences adopting a GT-C fold. We define three major clusters within the GT-C fold families. The GT-C0 cluster families have higher layer 3 average CAM values toward the N-terminal helices, which most likely is enough to separate them from GT-C1 and GT-C2. In contrast, GT-C1 includes families that are generally shorter in sequence length with little to no contiguous loop segments. The layer 3 average CAM values for these families stay high

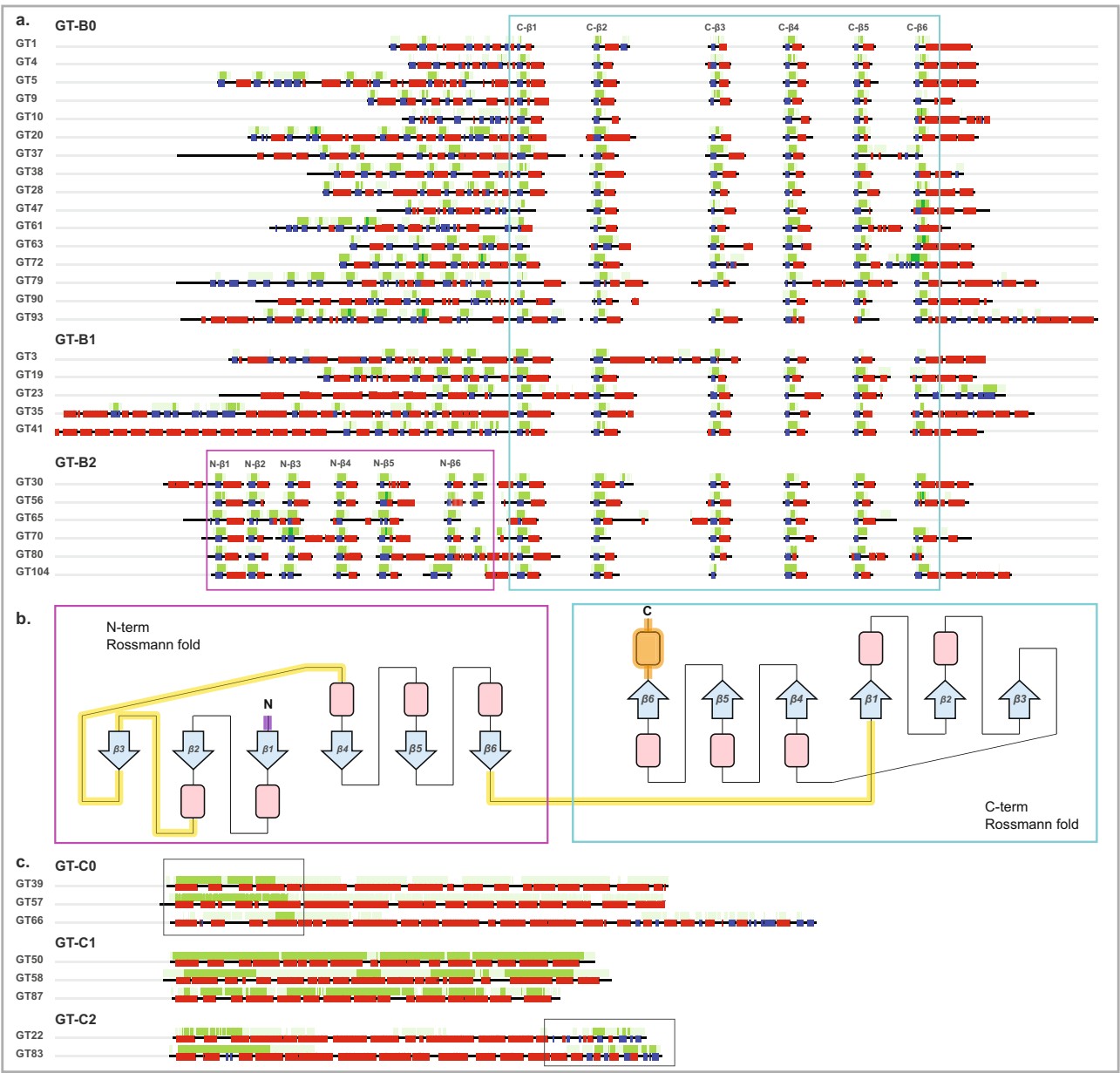

**Fig. 4 CAM maps for the different GT-B and GT-C fold clusters highlight their respective conserved cores. a** Consensus secondary structure (blue: beta-sheets; red: helices; green: loops) and average CAM values (higher intensity of green corresponds to a higher CAM value) from layer 2 are shown for all families belonging to the three GT-B fold clusters. These average values were generated from sequence alignments within each family. High CAM values within the cyan box point to the C-terminal Rossmann fold conserved across all GT-B fold members and the magenta box points to the N-terminal Rossmann fold conserved in GT-B2. **b** A topological representation of the conserved features of GT-B. The conserved C-terminal Rossmann-like fold region is shown in the cyan box. The N-terminal Rossmann fold, which is most conserved in members of GT-B2 cluster is shown in the magenta box. Conserved beta-sheets are shown as blue arrows with labels, and alpha-helices are shown as red boxes. Loop regions that have the most variability across families are indicated by yellow lines. Purple N-terminal loop and orange C-terminal helix indicate the presence of variable secondary structures preceding the N-terminal and following the C-terminal Rossmann fold, respectively. **c** Consensus secondary structure and average CAM values from layer 3 for GT-C families from clusters GT-C0, GT-C1, and GT-C2. Boxes indicate regions with higher average layer 3 CAM values for GT-C0 and GT-C2 in the N-terminal and the C-terminal regions, respectively. For GT-C1, layer 3 CAM is high throughout the full length of the sequences. Source data are provided as a Source Data file.

throughout the length of the sequences. Moreover, all members of the three families in GT-C1 are mannosyltransferases (PigM family GT50, Alg3 family GT58, and bacterial pimE of GT87), with PigM and Alg3 also known to share detectable sequence similarity[38]. Finally, GT-C2 members are distinct from other GT-C clusters in the C-terminal region where they share a distinct region with an α/β/α arrangement. This region has been identified as a periplasmic domain in a bacterial aminoarabinose

transferase ArnT of the GT83 family[39], which could interact with the donor substrate. Outside of the GT-C2 cluster, only GT66 family members (oligosaccharyltransferases) in GT-C0 have a similar extended C-terminal domain (Fig. 4c).

**Identifying families with novel GT folds using the convolutional autoencoder model.** While our CNN model could

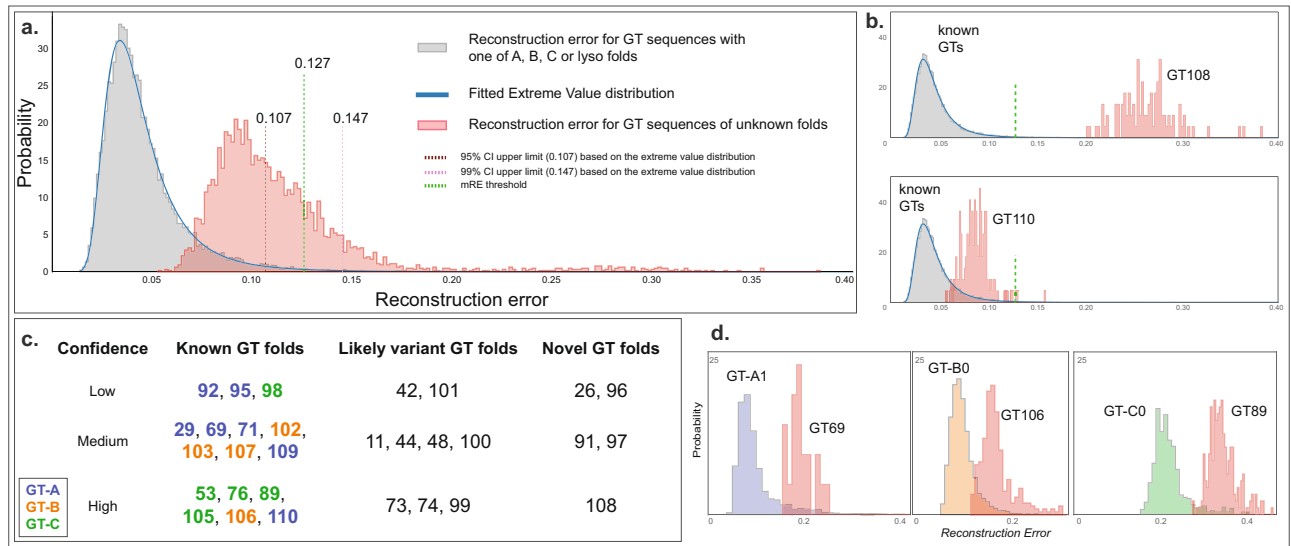

**Fig. 5 Fold prediction in GT-u families.** Reconstruction error (RE) for the known GT fold families are shown in gray and GT-u in red. **a** An extreme value distribution is fitted into RE for known fold to calculate a 95% and a 99% CI (upper limits in brown and pink dotted lines, respectively). The midpoint threshold RE at 0.127 ("Methods") is marked with green dotted lines. **b** As examples, RE for unknown fold families GT108 (upper panel) and GT110 (lower panel) are shown where RE for GT108 is very high and thus predicted to have a novel fold with high confidence. In contrast, the RE for GT110 is low and close to known fold families, thus predicted to have a known GT fold. **c** Chart showing the fold prediction results for 30 GT-u families with unknown folds. Family names are placed based on their likelihood of adopting a novel fold and the confidence in that evaluation and are colored based on their assigned fold types. **d** RE for three GT-u families predicted to have a known GT fold is plotted alongside RE for their predicted fold cluster with the highest fold assignment score (FAS). Left: GT69 versus GT-A1; middle: GT106 with GT-B0; right: GT89 with GT-C0. mRE and FAS scores for all the GT-u families are provided in Supplementary Table 5. Source data are provided as a Source Data file.

successfully distinguish the known GT fold types, there are 30 CAZy GT families (GT-u) that could not be assigned to a known fold based on literature review for the standard CNN-attention workflow. We wanted to extend our model to analyze and predict the fold types for these unknown families. To this end, we extended our existing CNN model to build an autoencoder that allows the calculation of a reconstruction error (RE) for any given sequence ("Methods"). Sequences similar to the ones used in training (i.e., one of the known folds) would have a low RE whereas novel fold sequences would have a large RE. Figure 5a shows the distribution of RE for sequences with known (GT-A, -B, -C, and -lyso in gray) and unknown folds (in red). There is a clear separation with the unknown fold sequences having a higher RE.

To statistically evaluate which GT-u families have a significantly higher RE than the known folds, we first fitted an extreme value distribution to our training data (RE from sequences with known folds) to calculate 95% and 99% confidence intervals (CI). We then compare a median RE value (mRE) for each GT-u family against these CI to make fold predictions. However, we note that the peak for unknown RE distribution falls within the 95% CI (below 0.107, Fig. 5a), suggesting that a majority of GT-u sequences adopt one of the known folds. For families that are predicted to adopt a known fold, we also wanted to identify their closest known fold type. To achieve this, we further built 9 autoencoder models for each of the two GT-A, three GT-B, three GT-C, and one GT-lyso clusters and calculated RE. Due to the low number of sequences in these cluster-specific models, instead of fitting an extreme value distribution, we used a fold assignment score (FAS, one for each subcluster totaling nine FAS scorers for each GT-u family), to evaluate the best match for each of the GT-u families. We derived the FAS score as a metric that provides a quantitative measure of how diverse any GT-u family is from all the known folds. This is done by comparing the RE values for a GT-u family against the median RE values of a cluster within the

known folds. This comparison is normalized using the RE values obtained for sequences that do not fall within that same cluster or that same fold such that a positive and high FAS score indicates similarity between a given cluster and the GT-u family being compared. The equation used to calculate this metric is provided in the Methods section along with additional details. Finally, based on the mRE and FAS scores for each GT-u family, we predict their fold status with varying degrees of confidence using the criteria described in "Methods" (Fig. 5, Supplementary Fig. 8, and Supplementary Table 5). In short, GT-u families with mRE less than a threshold of 0.127 (midpoint between the 95% threshold of 0.107 and the 99% threshold 0.147, details in "Methods") and a positive FAS score are designated as a known fold type and assigned to the fold corresponding to the highest FAS score. If a GT-u family has an mRE less than 0.127 but no positive FAS scores, they are designated as a variant fold type. Finally, families with mRE greater than 0.127 are designated as novel fold types.

Five families have very high mRE (larger than 0.127), and are predicted to adopt novel GT folds (Fig. 5c, Supplementary Fig. 8, and Supplementary Table 5). The dual-activity mannosyltransferase/phosphorylases of family GT108 have the highest mRE (0.281) and have indeed been shown to adopt a unique five-bladed β-propeller fold that is completely different from the four GT folds[40]. Another family predicted to have a novel fold, GT26, has a single representative crystal structure for a membrane-associated GT TagA, from a bacteria *T. italicus*, which also adopts a novel fold[41]. Here, we predict three additional families, the fungal β-1,2-mannosyltransferases Bmt/Wry (GT91), plant peptidyl serine α-galactosyltransferases Sgt (GT96), and bacterial α-2,6-sialyltransferases (GT97), that likely adopt novel GT folds as well.

Using the mRE and the FAS scores, we assign seven GT-u families as having a GT-A type fold (Supplementary Table 5). Out of these, the GT29 mammalian sialyltransferases have been

shown to adopt a modified GT-A fold with different orientations of the beta-sheets in the Rossmann fold while conserving the overall Rossmann-fold scaffold and specific sialyl motifs[42]. The human glycolipid glycosylphosphatidylinositol β-1,4-N-acetylga-lactosaminyltransferase PGAP4 (GT109) has also been predicted to adopt a GT-A fold with transmembrane domain insertions[43]. In line with this study, the GT109 family is predicted to have a GT-A fold with medium confidence. Our analysis further adds the α-1,3-mannosyltransferases (GT69) to the GT-A fold families with medium confidence. In addition, we predict that the α-mannosyltransferases Mnn (GT71), the plant GalS galactan synthases and other members of the GT92 family, members of the GT95 family (hydroxyproline β-L-arabinofuranosyltransferase HPATs), and the β-1,4-xylosyltransferases (Rxylt1/TMEM5) of the GT110 family also adopt folds that are similar to the GT-A type fold.

We also identify four GT-u families that most likely adopt the GT-B fold (Fig. 5c and Supplementary Table 5). This includes the bacterial α-1,3-L-rhamnosyltransferase (GT102), the bacterial O-antigen-polysaccharide β-1,4-N-acetylglucosaminyltransferases (GT103), the GT106 family of plant rhamnogalacturonan I 4-α-rhamnosyltransferases, and the GT107 family of KDO trans-ferases. Similarly, five families have the highest positive FAS score against GT-C clusters and are predicted to adopt the GT-C fold (Supplementary Table 5). In agreement with our predictions, cryo-EM-based structures of representative bacterial Embs of the GT53 family have revealed a GT-C fold[44] and recent structural predictions on the human TMTCs of family GT105 have suggested that they adopt a GT-C fold[45]. In addition to these two families, we predict that PigVs (GT76), bacterial arabinofur-anosyltransferases AftBs (GT89), and dpy-19 mannosyltrans-ferases (GT98) also adopt a GT-C fold. In addition, all of these five families utilize lipid-linked sugar donor substrates similar to other known GT-C fold enzymes[29,44,46].

The remaining nine families have a negative FAS score for all the GT-A, -B, -C, and -lyso clusters and thus are not assigned a specific fold type. However, since they have an mRE below 0.127, these families are predicted to adopt a variant of the existing fold types rather than a novel fold type. Among them, families like the bimodular dual β-glucosyltransferases of GT101 and the multi-modular bacterial β-KDO transferases of the GT99 family have representative crystal structures[47,48] revealing that they adopt unique folds consisting of the Rossmann-fold scaffold with the latter forming a variant of the GT-B fold type. The bacterial toxin glucosyltransferases of the GT44 family have also been shown to adopt a slightly modified structure highly similar to a GT-A fold[49,50]. Bacterial Csts from the GT42 family also have been shown to adopt a variant of the GT-A fold type that is highly similar to the GT29 sialyltransferases with both families conserv-ing the sialyl motifs[51]. Yet, while GT29 scores higher against the GT-A1 cluster, GT42 does not and is correctly classified as a variant fold type suggesting key differences in other regions of the GT-A core. Here, we add variant fold predictions for the GT11 (fucosyltransferases), GT48 (glucan synthases), GT73 (bacterial KDO transferases), GT74 (includes few α-1,2-L-fucosyltrans-ferases), and GT100 (bacterial sialyltransferases) families. Addi-tional details of these predictions are provided in "Methods", and the results are summarized in Supplementary Table 5.

## Discussion

It has been well-established that the structural folds of GTs, much like in many other large protein families, are far more conserved than primary sequence[5,6]. The functional diversification of GTs through extensive sequence variation and insertion of variable loops and disordered regions presents a major challenge for broad

sequence or structural classification using alignment-based approaches. This inability to create a larger framework of GT structural classification has impeded understanding of the evo-lutionary relationships among GTs during the expansion of gly-can diversity in all domains of life[52,53]. Although GTs primarily adopt three major fold types, each has its own distinct features. GT-A and GT-B enzymes employ single or paired Rossmann folds, respectively, for donor and acceptor binding during cata-lysis. Less is known about GT-C fold enzymes that employ dis-tinct features composed of multiple transmembrane helical domains. Identifying and distinguishing the GT-A and GT-B folds in the absence of solved structures is quite challenging and nontrivial, more so when the starting fold type is not known as is the case for multiple GT-u families. To overcome these challenges and produce reliable fold predictions, we use a CNN-attention-based deep-learning model that implements a completely alignment-free approach relying simply on secondary structure patterns to classify all GT families into either the known fold types or predict novel fold types. As far as we know, this is the first attempt at utilizing this simple coarse-grained, dependable form of input for analyzing such a large group of enzymes using deep learning. As such, our proposed method provides a novel approach for fold classification by using the secondary structural features that can be useful in studying evolutionarily divergent families such as GTs. We successfully built a model that classified known folds and families with 96% and 77% accuracy, respec-tively. In addition, we focus the design of our model on inter-pretability, where each layer generates outputs in the form of CAM maps (Block 1), features for UMAP visualization (Block 2), and reconstruction errors (Block 3) for biological interpretation and understanding of the model.

By mapping the features learned by the model using UMAP, we identified clusters of families within the major fold type that were found to share distinct regions of similarity, as revealed by their CAM maps. Each of the two clusters within the GT-A fold include phylogenetically diverse families[3] yet each shares a unique set of secondary structural features within the hypervariable regions that distinguish the clusters and likely contribute to substrate recognition. Because such features shared by evolu-tionarily divergent families are difficult to detect through traditional phylogenetic approaches, the CNN model provides a valuable alternative tool for inferring such shared structural mechanisms. These new insights into the common secondary structural fea-tures can serve as valuable starting points for informed testing of hypotheses regarding GT-A fold evolution, enzymatic specificity, and function.

In the GT-B fold families, where previous attempts of sequence and structural alignments have proven difficult, we identify a central GT-B0 cluster that points to a limited conserved core in the C-terminal Rossmann fold, with insertions in the loop regions. We show that this conservation extends across the large and diverse GT-B0 cluster to other GT-B clusters as well (Fig. 4a and Supplementary Fig. 7). In the smaller GT-B2 cluster, CAM maps point to additional structural similarities in the N-terminal Rossmann domain within this cluster. By virtue of these shared features, we present an alignment of predicted secondary struc-tures across GT-B fold families providing a comparative basis for cross-cluster analyses (Fig. 4). Similarly, we identify a subset of GT-C fold families (GT-C1) consisting entirely of mannosyl-transferases where the CAM features extend throughout the entire length of the sequences (Fig. 4c).

More importantly, we deploy an autoencoder model using the features from the CNN-attention model to make reliable pre-dictions for GT-u families that are most likely to adopt a novel GT fold. The 16 GT-u families found to adopt known folds (Fig. 5c) provide a comparative basis for understanding their

functions and associations. On the other hand, five GT-u families are predicted to adopt novel folds. Three out of the five families (GT91, GT96, and GT97) do not have a representative crystal structure. Coincidentally, each of these three families is found in select taxonomic groups (fungi, plants, and bacteria, respectively) and has different functions. Moreover, out of the 12 families that are predicted to adopt variant folds, only 4 (GT42, 44, 99, and 101) have representative crystal structures, all of which point to unique structural adaptations and variations[47–51]. Our predictions for other families that lack representative structures provide informed targets for focused structural studies that could reveal divergent GT folds with different mechanisms and modes of regulation to expand the GT fold space and uncover unique aspects of GT function, regulation, and evolution.

We use a combination of metrics (RE, FAS, number of sequences) to assign confidence levels for our predictions providing researchers with meaningful metrics of reliability for guiding future efforts. These predictions are based on the family level and utilize secondary structure predictions on a large number of sequences from each family, thus providing robust results. However, interpretations for families such as GT78 (A fold), GT18 (B fold), GT103, or GT97 (novel fold) with very few unique sequences should be done with caution.

Finally, our approach employs a simple training dataset that is straightforward to prepare and is surprisingly adaptable for understanding fold diversity in any large protein family. Indeed, preliminary application of our model to the classification of protein kinases demonstrates that the features learned by the model can successfully distinguish the protein kinase fold sequences from non-protein kinase fold sequences with ~99% accuracy. Furthermore, similar to GTs, the model also separate the major kinase groups with 83% accuracy (Supplementary Fig. 9), suggesting that the model is capable of finding small structural differences to distinguish protein kinases at the group level. Contrary to most "black box" deep-learning models, the output of this workflow is a highly interpretable deep-learning model that generates accurate fold predictions with quantitative outputs that provide meaningful biological insights without the need for primary sequence or structural alignment. Thus, the approach adds a powerful tool to the repertoire for computational and evolutionary analyses of large protein families.

## Methods

### Data collection and preprocessing

*Sequence retrieval and secondary structure prediction.* We retrieved GenBank[54] IDs for GT sequences from the CAZy database (accessed May 4, 2020). Sequences for these IDs were then collected from the NCBI GenBank database. These sequences were first filtered using the USEARCH[55] method to remove sequences that share >60% similarity for large GT families (with >5000 members listed in CAZy), 80% similarity for GT families with 500–5000 members, and 95% similarity for smaller GT families (with <500 members) to balance the number of sequences across families and to avoid overfitting. We predicted the secondary structures of our filtered dataset of 44,620 sequences using NetSurfP2.0[24]. NetSurfP predicts both three-state and eight-state secondary structures based on DSSP definitions[56]. Here, we only use the three-state predictions as input features since these are reported with higher accuracy. In addition, we make our predictions on the family level, which accounts for persistent secondary structure predictions in multiple closely related sequences from the same family that makes our method robust to small inconsistencies in secondary structure prediction for individual sequences.

*Sequence length filtering.* To allow batch training for neural networks, these sequences were padded to a consistent length of 798. We set this threshold by modeling the distribution of GT-A-, B-, C-, and -lyso sequence lengths to a Gaussian distribution and setting our maximum length cutoff at $\mu + 3\sigma$. However, for a subset of sequences that extend beyond 798 amino acids, we eliminated sequences flanking the GT domain through domain mapping via Batch CD-search[57]. Sequences with multiple GT domains were labeled separately and treated as different sequences. Sequences lacking an annotated GT domain or with an annotated GT domain longer than 798 amino acids were removed. Sequences shorter than 798 amino acids were padded to this length by adding a vector [0,0,0]

for each padded position. Our final padded dataset contained 12,316 GT-A, 20,397 GT-B, 1518 GT-C, 5482 GT-lyso, and 4258 GT-unknowns where each sequence is represented by a 798X3 matrix of secondary structure predictions and padding.

*Data augmentation for balancing datasets across families.* Skewed datasets can hinder the convergence of neural networks and negatively impact generalization. To mitigate this issue, we balanced our training dataset using data augmentation. Our data augmentation procedure randomly changes 5% of secondary structure positions to coil/loop, excluding the padding region. This procedure can sometimes produce no changes, such as if only coil/loop positions are randomly chosen. In these cases, the procedure is repeated until at least one change is made. To generate our balanced training set, we used this data augmentation strategy to increase the number of sequences to 2000 sequences for each of the GT-A, GT-B, and GT-C fold families. For the single GT51 family of GT-lyso fold, we randomly selected only 5000 sequences after performing the sequence similarity filtering. For two families with a very large number of sequences, GT2 and GT4, we selected 2000 divergent representatives from each family. This balanced training set was generated once and reused for parameter optimization unless otherwise indicated.

**CNN model for fold and family classification.** The model architecture involves an attention-aided deep CNN model with six blocks. The first three blocks (Block 1, Fig. 1) sequentially use a one-dimensional convolutional layer, followed by a pooling layer and a batch normalization layer. This feeds into an attention[27] layer that performs a refinement of the generated feature maps. The convolution kernel sizes were set to 3, 7, and 15 with kernel numbers set to 256, 512, and 512, respectively, for the first three blocks. Since pooling operations lead to a loss of spatial information for the feature maps, such an operation is not applied on any of the first three layers, thus enabling mapping of the attention maps back to the sequence for interpretation.

The feature maps from these three layers are downsampled using a global average pooling layer and then passed through three additional blocks before making the final prediction. In contrast to the first three layers that carry spatial information, these three layers (Block 2, Fig. 1) use global max pooling operations that compute a single maximum value for each of the input channels, thus providing a single linearly independent representation for each sequence, regardless of sequence length. These representations can be transformed into high-dimensional vectors which can then be used by downstream clustering algorithms.

For the multitasking of fold and family classification, two separate fully connected layers were added with dropout. The model was trained on a single NVIDIA RTX 2080Ti graphic card for 6 h. The dropout rate was set to 0.5 during training. Adam optimizers with a learning rate of 1e-4 and weight decay with a rate of 1e-5 were deployed during training.

*Comparison to other methods.* For the Transformer model, we used ESM-1b[17] to generate embeddings for each sequence. Mean values were calculated across all positions to generate one vector of shape (1280,1) for all sequences. We used UMAP for dimensionality reduction and visualization. The embedding vectors were also taken as input by the gradient boosting decision tree (GBDT) model for fold- and family-level classification. For the LSTM model, we used a single-layer biLSTM model with 64 hidden units. For the ProtCNN model, we followed the original code implementation with two residual blocks. Overall, our model demonstrated advantages in three aspects: (1) better overall accuracy, (2) ability to classify GT-u families, and (3) better interpretability. A detailed comparison can be found in Supplementary Table 3. For the comparison with the HHsearch method, HHM profiles were generated along with secondary structure predictions for all the GT families of known folds and used to build a HHsearch database. Multiple sequence alignments of sequences within each GT-u family were then used as queries to search the known GT fold database. The top three results with an -value less than 1e-2 were selected for fold assignment for any given GT-u family. The results are provided in Supplementary Table 2.

**Autoencoder framework for identification of novel fold GT-u families.** We adopted a recent advancement in the machine-learning field named open set recognition[28] to extend the trained classifier's ability to distinguish an unseen pattern of secondary structure from the seen dataset of known GT folds. In application, this framework is targeted to real-world scenarios where new classes (unknown classes), unseen during training, appear in the testing phase and requires the classifier to not only accurately classify seen classes but also effectively deal with unseen classes in testing[28]. This translates well to our problem of distinguishing GT families that most likely adopt a previously unseen fold which is considerably different from the GT-A-, B-, C-, and -lyso folds that the model is trained on, while efficiently recognizing families that could adopt one of these known folds. We propose a CNN-based autoencoder framework to accomplish this task, which is capable of reconstructing the known GT folds that it has learned on but unable to do so if a given sequence is quite different, resulting in a high reconstruction error.

The autoencoder (Block 3, Fig. 1) comprises two parts: an encoder and a decoder. The encoder reused Block 1 of the CNN model trained on GT-A, -B, -C, and -lyso as a general feature extractor. Then, a mirror structure of the CNN model that includes multiple deconvolution operations and instance normalization is

connected to the encoder to generate a reconstruction of the inputs. The similarity between the seen and unseen classes is measured by calculating a reconstruction error (RE) of the input samples (Fig. 5a). A modified loss function was proposed to calculate RE in order to omit the effects of padding regions in the reconstruction of sequences as follows:

$$\text{Masked MSE} = \frac{1}{n-2p}\sum_{1+p}^{n-p}(Y-\hat{Y})^2 \qquad (1)$$

where $p$ is the padding length at both ends of the sequence, $n$ is the sequence length, $Y$ is secondary structure input, $\hat{Y}$ is the predicted secondary structure output.

In addition to this main autoencoder model, nine additional autoencoder models were built with the same architecture but trained separately on the nine clusters of GT folds: two GT-A, three GT-B, three GT-C, and one GT-lyso clusters. RE against each of these clusters were used to derive a fold assignment score (FAS) that was used as a measure to indicate which known fold a given GT family would adopt, if it was predicted to adopt a known fold. The FAS score was calculated using the following equation:

$$\text{FAS}_{ab} = \left(\frac{(\text{OOC}_b - \text{RE}_a) \times (\text{OOF}_b - \text{OOC}_b)}{\text{OOF}_b - \text{RE}_b} - \text{thres}\right) \times 100 \qquad (2)$$

where $\text{FAS}_{ab}$ is the fold assignment score for GT-u family $a$ against cluster $b$, $\text{RE}_a$ is the median RE for sequences in family $a$, $\text{RE}_b$ is the median RE for sequences in cluster $b$, $\text{OOC}_b$ is the average RE for sequences with the same fold as sequences in cluster $b$ but are not grouped in cluster $b$ (called out of cluster (OOC)), $\text{OOF}_b$ is the average RE for sequences from a different fold than sequences in cluster $b$ (called out of fold (OOF)), and thres is a threshold score for fold prediction. Since GT-lyso had only one cluster, 20% of sequences were left out of training, unseen by the model and used to calculate $\text{OOC}_{\text{lyso}}$. The threshold thres was set to 0.014 based on the RE distributions to account for the differences in $\text{RE}_{\text{clusters}}$ across different clusters. Distributions of RE, OOC and OOF for each of the nine clusters are provided in Supplementary Fig. 10. In all of these autoencoder models, a smaller set of 200 sequences each for the GT2 and GT4 families were used so that these models did not overfit the two large GT families. For all other GT families, all non-augmented sequences used in training the CNN model were used.

### Model interpretation

*Structural mapping of layer-wise activation maps*. To fully understand how the CNN-attention model classifies GT fold types, we analyze feature maps generated from Blocks 1 and 2 using two different methods:

1. For all three layers of Block 1, we rely on making weakly supervised class-specific localization through a label-guided method named class-specific activation mapping using Grad-CAM[30] (CAM) that uses gradient descent to generate feature maps that target-specific families. The attention layers inserted in Block 1 further enhance these activation values. These Grad-CAM results were used to generate activation maps that conserve spatial information and can be mapped back into the sequence to identify the most contributing secondary structure and sequence regions for fold classification and thus represent the core conserved features.

2. For the three layers of Block 2, we generate saliency maps that highlight activations by extracting the feature map values. These maps do not conserve spatial information but are used to generate representation vectors that are then subjected to dimensionality reduction using UMAP[32] to generate manifolds for visualization and clustering of the known GT fold types. To identify the major clusters within GT fold types, we clustered the 2D UMAP projections using the GMM algorithm[35]. UMAP was performed with multiple sets of parameters to find families that most consistently grouped together (Supplementary Fig. 3). For the GT2 and GT4 families that have a very high number of sequences, we used the same 200 representative sequences used in the autoencoder models for UMAP visualization. When implementing the GMM algorithm, an appropriate cutoff for the GMM score was selected independently for each fold type in order to generate clusters robust to changes in parameters of UMAP.

*Evaluation of the reconstruction error to identify novel fold type families*. Since the RE for most training sequences would be very low, the RE distribution for the training data from the main autoencoder was first fitted to an extreme value distribution using the scipy[58] package. This was then used to evaluate a 95% CI and a 99% CI. Median RE (mRE) calculated for each GT-u family was then compared to these two CI limits to statistically evaluate their likelihood of adopting a novel fold. In addition, the FAS scores were used for fold assignments of the families predicted to adopt known folds. A positive FAS score indicates that the RE value for that family scores better against a given cluster than RE values for families that are from a different cluster or a different fold, suggesting similarity between that family and cluster. Thus, an appropriate mRE threshold should separate all families that have a positive FAS scores from families with all negative FAS scores. The value 0.127 that marks the midpoint for the interval between the upper limits of the

95% and the 99% CI (0.107 and 0.147, respectively) was found to be an appropriate cutoff and used as a threshold for predicting GT-u families that adopt a novel fold (higher mRE than 0.127) or a variant of the known folds (mRE lower than 0.127). Further evaluation of the prediction was done using the FAS scores as follows:

1. For families with mRE lower than 0.107 (95% CI), the highest FAS score was always positive, and the GT-u family was assigned to the fold with the highest FAS score. Families with an mRE score lower than 0.1 and an FAS score higher than 1 were considered high confidence, while others were considered medium confidence.

2. For families with mRE between 0.107 and 0.127, if the FAS scores were positive, they were assigned to the fold with the highest FAS scores with low confidence. If the FAS scores were all negative, those families were assigned as variants of known folds. Higher mRE scores corresponded to an increase in confidence.

3. For families with mRE higher than 0.127, FAS scores were always negative. These families were designated as novel fold types with an increase in mRE scores corresponding to higher confidence.

The code and related datasets for conducting these analyses are made available at https://doi.org/10.5281/zenodo.5173136[59].

**Reporting summary**. Further information on research design is available in the Nature Research Reporting Summary linked to this article.

## Data availability

For the sequences used in this study, the IDs were collected from the CAZy database (http://www.cazy.org/GlycosylTransferases.html) and using those IDs, the sequences were obtained from the NCBI database (https://www.ncbi.nlm.nih.gov/protein/) using batch Entrez (https://www.ncbi.nlm.nih.gov/sites/batchentrez). All the sequences and their secondary structure predictions that were used for training and testing both the CNN-attention and the autoencoder models have been made available through https://github.com/esbgkannan/GT-CNN. A list of GenBank IDs for all the GT sequences used in this study is also provided in Supplementary Dataset 1. The sequence and secondary structure prediction data, along with the code used in this study are available in Zenodo[59]. Source data are provided with this paper.

## Code availability

The code used to train and implement the deep-learning framework described here was written in Python 3.7 and is available as Jupyter notebooks, along with detailed requirements and steps, from https://github.com/esbgkannan/GT-CNN. The published version of the code with the manuscript is available at https://doi.org/10.5281/zenodo.5173136[59].

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

## Acknowledgements

Funding for N.K. from NIH R01GM130915 and R35 GM139656 is acknowledged. R.T. was also supported by T32 GM107004.

## Author contributions

R.T., Z.Z., and N.K. conceived and designed the project. Z.Z. and S.L. developed the deep-learning models. R.T. and Z.Z. wrote the scripts, performed the analyses, and drafted the manuscript. W.Y. assisted in designing the deep-learning model, data analysis, and writing scripts. S.L., K.W.M., and N.K. supervised the project and revised the manuscript. All authors reviewed and approved the final manuscript.

## Competing interests

The authors declare no competing interests.
