## [Peer Review File · Nature Communications]

Mapping the glycosyltransferase fold landscape using interpretable deep learningReviewers' Comments:

Reviewer #1:

Remarks to the Author:

This manuscript presents a deep learning approach for classifying the folds of Glycosyltransferases (GTs). This is an interesting and difficult problem because of the significant sequence divergence between GTs. The authors present a convolutional neural network with attention (CNN-attention) that successfully classifies the folds of GTs based just on their predicted secondary structure. Importantly, this approach does not require a sequence alignment, allowing it to handle the enormous sequence diversity of GTs. I really enjoyed the manuscript.

My main request is that the authors provide some more of the intuition for their architectural choices in the main text, as too much of that information is currently in the Methods section. For example, what is a CAM and how does it enable interpretation of the network? What is an FAS score and how are they interpreted?

A few minor points:

How did the CNN suggest some sequences couldn't be classified?

If the CNNs already suggested some sequences can't be classified, what is the autoencoder RE analysis really adding?

How were the mRE cutoffs for classifying folds as novel chosen?

This is Greg Bowman, at Washington University.

Reviewer #2:

Remarks to the Author:

The authors trained a NN (using CNN and attention layers) for GT fold classification given predicted secondary structure. The authors also train an autoencoder for each GT-fold, and use the reconstruction cost as a fold assignment score, allowing for identification of novel GT-u families, if none of the autoencoders are able to reconstruct these. An impressive amount of work went into this manuscript, and I think it has the potential to transform how fold classification is done. But... without additional work showing how it compares to other methods and demonstration on other protein families, I'm a bit concerned about the broader impact of the work.

Big picture/Major concerns:

- A new method was developed specifically for one protein family and tested on the same protein family. Though the authors did take care to split the single protein family into training and test set, it is still concerning. I would be convinced if the authors applied the protocol on a completely different dataset, demonstrating "applicable to other broad, heterogeneous protein families where challenges in primary sequence alignment approaches have hindered analysis of fold classification and evolutionary relationships."

- Though the amount of deep-learning incorporated into the study is quite impressive, it is not clear if it is actually needed. Without comparison to traditional methods, it's not clear how impressive the results really are. What if one was to simply count the number of different secondary structure elements for each sequence, and clustered the sequences based on this. Or used the Unirep representation of sequences <<https://www.ncbi.nlm.nih.gov/pmc/articles/PMC7067682/>> which was shown to learn secondary structure, and is "alignment-free" how would this compare? Or how about Transformer models trained on all of UniProt, which also use attention maps and have been shown to learn contacts (representing secondary and tertiary structure). <<https://arxiv.org/abs/2006.15222>> .

Surely, the representation learned here could be used to classify GTs. Finally, ProtCNN model specifically designed for classification: <<https://www.biorxiv.org/content/10.1101/626507v4.full>>

- Though the authors do mention issues in constructing alignments, it does seem that alignments of each sub-type could be constructed? Or what if one was to simply align the strings of predicted secondary structure elements, and used that as the metric? Methods like HHsearch (and other fold-recognition methods) construct an HMM for a cluster of related sequences, also incorporate predicted secondary structure for remote homology search/clustering. It seemed one could have simply constructed an HMM+SS for each subfamily/classification of proteins and then used the provided score metric for fold assignment for new members.

Minor concerns:

- Page 18, line 394: "filtered to sequence similarities of 60-95%". This is not clear. Filtered relative to some reference sequence? filtered to remove redundancy (where no two sequences share more than 60 or ... or 95% identity)? What method was used for filtering?

- Page 20, line 422: "upsampled to 200 sequences". Not sure what this means. Did you repeat the same sequence multiple times?

Reviewer #3:

Remarks to the Author:

The authors employ a deep learning model to predict structural folds for glycosyltransferases (GTs) from primary sequences. GTs comprise an enormous class of enzymes that have been classified into 114 CAZy families based on sequence similarity. To date three major structural folds have been identified for glycosyltransferases; GT-A, GT-B and GT-C as well as a lysozyme type fold for family 51 peptidoglycan polymerases. The authors have previously identified a common scaffold for GT-A fold enzymes using sequence mining methods, however, these methods are not applicable to the other fold types (Taujale et al, 2020). In the present study, they achieve an accuracy of 96% based on fold prediction and 77% on family classification in a representative set of sequences. The GT landscape showed distinct clusters for the respective GT folds. Two substructures in the GT-A fold and three for GT-B and GT-C folds were identified in the clusters. The structural basis for the separations in the two GT-A fold substructures are attributed differences in helices in a hypervariable region and at the C-terminus. Patterns for the three GT-B and GT-C substructures were also determined including conserved cores. Thirty CAZy families with unknown folds were predicted to adopt known folds or their variants including five that were predicted to have novel folds. Two of these have been structurally confirmed while the three others are members of CAZy family 91, 96 and 97.

I am enthusiastic about the computational methods developed and their applications to mine sequence data. Patterns were identified for conserved GT fold clusters and subtypes within these clusters and novel fold types were predicted. To my knowledge global analyses are unprecedented for the GT-B and GT-C folds. My enthusiasm is tempered by a lack of insight into GT evolution or function although pattern identification is a step in determining evolutionary relationships

Reviewer #1: (Expertise: DL for protein structure-function

mapping): Summary:

This manuscript presents a deep learning approach for classifying the folds of Glycosyltransferases (GTs). This is an interesting and difficult problem because of the significant sequence divergence between GTs. The authors present a convolutional neural network with attention (CNN-attention) that successfully classifies the folds of GTs based just on their predicted secondary structure. Importantly, this approach does not require a sequence alignment, allowing it to handle the enormous sequence diversity of GTs. I really enjoyed the manuscript. My main request is that the authors provide some more of the intuition for their architectural choices in the main text, as too much of that information is currently in the Methods section. For example,

Comment #1: What is a CAM and how does it enable interpretation of the network?

Response: We thank the reviewer for the positive comments. CAM stands for Class-specific Activation Map (CAM) which is a projection of the feature space on to the primary sequence. Such projections enable interpretability of the model by highlighting key sequence and structural features contributing to the classification. We have added additional text in the revised version (lines 110 – 116) to provide explanation of CAM and how we're using it to interpret the results of our GT-CNN model.

Comment #2: What is an FAS score and how are they interpreted?

Response: Fold Assignment Score (FAS) is a comparative metric that provides a quantitative measure of how similar (or different) a GT family of unknown fold is to one of the known fold types. A high positive FAS score for a known fold indicates a high similarity to that fold type. We have revised the manuscript to provide additional explanation and interpretation of the FAS scores (lines 285 – 291). Additionally, in the Methods section we provide details for FAS score calculation and interpretation.

A few minor points:

Comment #3: How did the CNN suggest some sequences couldn't be classified? If the CNNs already suggested some sequences can't be classified, what is the autoencoder RE analysis really adding?

Response: The CNN model is trained on GTA, B, C and Lyso folds and while the model classifies sequences within these fold classes with high accuracy, sequences adopting novel folds are incorrectly forced into one of the four classes (GTA, B, C or Lyso). To address this issue, we developed the autoencoder model based on open set recognition. The autoencoder models were trained with respect to each of the predefined subclusters of GTA, B, C and Lyso. Subsequently, different GT-u families were compared using RE analysis. This

enables us to extend the CNN-Attention model to classify GT-u families without explicitly training on them. We have now clarified this in the revised version.

Comment #4: How were the mRE cutoffs for classifying folds as novel chosen?

Response: The mRE cutoffs were chosen based on the statistical modeling of the RE distribution. We fitted an extreme value distribution to the RE values of known fold sequences to obtain a 95% and a 99% Confidence Interval. The upper limits for these intervals were at 0.107 and 0.147 respectively. Most GT-u families scored within these limits. We used the unbiased mid-point between these two extremes at value 0.127 as the cutoff to designate their fold status as known, novel or variants of known folds. Our predictions using these cutoffs are supported by GT-u families with some literature support describing them as likely known, novel or variants of known folds.

Reviewer #2: (Expertise: DL/ML for protein structural prediction):

Summary: The authors trained a NN (using CNN and attention layers) for GT fold classification given predicted secondary structure. The authors also train an autoencoder for each GT-fold, and use the reconstruction cost as a fold assignment score, allowing for identification of novel GT-u families, if none of the autoencoders are able to reconstruct these. An impressive amount of work went into this manuscript, and I think it has the potential to transform how fold classification is done. But... without additional work showing how it compares to other methods and demonstration on other protein families, I'm a bit concerned about the broader impact of the work.

Response: We thank the reviewer for acknowledging the volume of work that went into this manuscript and the potentially transformative nature of the work for fold classification

Big picture/Major concerns:

Comment #5: A new method was developed specifically for one protein family and tested on the same protein family. Though the authors did take care to split the single protein family into training and test set, it is still concerning. I would be convinced if the authors applied the protocol on a completely different dataset, demonstrating "applicable to other broad, heterogeneous protein families where challenges in primary sequence alignment approaches have hindered analysis of fold classification and evolutionary relationships."

Response: We thank the reviewer for this comment. Indeed, ongoing work in the lab is focused on extending the model for other protein classes such as protein kinases and ion-channels. To illustrate the applicability of our model to other protein superfamilies, we provide some preliminary results from an ongoing project to characterize the protein kinase fold superfamily. We generated secondary structure representations on 11675 kinase sequences curated in our lab and conducted fold prediction following the same workflow described in the manuscript for the GT family. The model classified protein kinase fold sequences from non protein kinase fold sequences with nearly 99% accuracy and separated

major protein kinase groups with 83% accuracy (Supplementary Figure 9). The classification of kinase groups and families is particularly interesting because the differences between protein kinase groups is typically defined by primary sequence information (which was not provided to our model). This suggests that our model is capable of finding small structural differences which can distinguish protein kinases at the group level. While we have included a supplementary figure (Supp Fig 9) and a brief discussion demonstrating the generalizability of our model (lines 418-424), for the current study, however, we would like to focus on the GT super-family for the following reasons. First, it represents one of the most challenging protein super-families to classify using traditional bioinformatics approaches. Second, focusing on the GT superfamily allows us to highlight the interpretable aspect of the model by describing sequence and structural features contributing to fold classification based on Class Activation Maps. Third, the GT-B and GT-C folds form an independent test set for our model and enables the first systematic classification of these fold classes. Finally, our prior experience in the classification of the GT-A fold class using traditional bioinformatic approaches enables a direct comparison with the deep learning model and makes our findings broadly applicable to the glycobiology community.

Comment #6: Though the amount of deep-learning incorporated into the study is quite impressive, it is not clear if it is actually needed. Without comparison to traditional methods, it's not clear how impressive the results really are. What if one was to simply count the number of different secondary structure elements for each sequence, and clustered the sequences based on this. Or used the Unirep representation of sequences <https://www.ncbi.nlm.nih.gov/pmc/articles/PMC7067682/> which was shown to learn secondary structure, and is "alignment-free" how would this compare? Or how about Transformer models trained on all of UniProt, which also use attention maps and have been shown to learn contacts (representing secondary and tertiary structure). <https://arxiv.org/abs/2006.15222> . Surely, the representation learned here could be used to classify GTs. Finally, ProtCNN model specifically designed for classification: <https://www.biorxiv.org/content/10.1101/626507v4.full>

Response: We thank the reviewer for this critical comment. We agree that all these methods present a viable alternative towards addressing this problem, though some of the recently proposed models such as the Transformer model are quite complex. Nevertheless, as requested by the reviewer, we have now performed extensive comparisons of our model with these methods. Our studies further support the value of our CNN-attention model in providing explainable classification of GT folds. In the revised version, we have added a new Suppl. Table 3 and Suppl. Fig 2 comparing our CNN-attention model with (1) A transformer embeddings + GBDT classifier, (2) a LSTM based classifier and (3) the ProtCNN model. Suppl. Table 3 compares the performance of these methods in terms of accuracy, interpretability and the ability to handle unknown fold classes. As shown in Suppl. Table 3, only the transformer model performs with comparable accuracy to the CNN-attention model and even then, lacks the ability to handle unknown fold classes and is much less interpretable. The clustering of UMAP projections generated using the transformer based model does not separate the GT folds well; the CAM maps cannot be obtained to delineate the conserved

core, nor can quantitative metrics be generated to differentiate novel or variant GT folds from the known fold types.

We have incorporated the comparison results in the main text (lines 101-108) and expanded on the comparison in the Methods section (lines 492 - 506). We believe these additions further strengthen the value of our CNN-attention model.

Comment #7: Though the authors do mention issues in constructing alignments, it does seem that alignments of each sub-type could be constructed? Or what if one was to simply align the strings of predicted secondary structure elements, and used that as the metric? Methods like HHsearch (and other fold-recognition methods) construct an HMM for a cluster of related sequences, also incorporate predicted secondary structure for remote homology search/clustering. It seemed one could have simply constructed an HMM+SS for each subfamily/classification of proteins and then used the provided score metric for fold assignment for new members.

Response: We thank the reviewer for pointing out these alternative methods. We would like to emphasize that the divergent nature of GT-B and GT-C fold families precludes the application of alignment-based methods for fold classification. Nevertheless, based on detectable sequence similarity within families, we generated secondary structure based HMMs for individual GT families and predicted folds of GT-u sequences based on profile similarity using HHsearch. HHsearch based predictions are compared to our deep learning framework (GT-CNN) in Supplementary Table 2 and described in the Results section (Lines 100-108) and Methods section (Lines 496-501). In summary, comparisons revealed that while HHsearch predicted folds of some GT-u families (GT26, GT53, GT108, GT109), it misclassified several (GT105, GT29, GT99, GT101). In addition, traditional methods like HHsearch still rely on generating accurate sequence alignments for comparisons, which can be a challenge. Furthermore, such methods rely on the choice of e-value cutoffs and cannot make a distinction between variant or novel fold types. Finally, the amount of biological insight that can be drawn from our CNN based method is highly valuable and novel and cannot be obtained from these methods.

Minor concerns:

Comment #8: Page 18, line 394: "filtered to sequence similarities of 60-95%". This is not clear. Filtered relative to some reference sequence? filtered to remove redundancy (where no two sequences share more than 60 or ... or 95% identity)? What method was used for filtering?

Response: We have clarified the sequence similarity based filtering step which was used to remove sequence redundancy. We now specifically mention the percent similarity cutoffs (60% for families with >5000 sequences, 80% for families with 500-5000 sequences and 95% for families with <500 sequences) used on GT families of different sizes. We've also added citation for the USEARCH tool that we used to perform the filtering (lines 434-437).

Comment #9: Page 20, line 422: "upsampled to 200 sequences". Not sure what this means. Did you repeat the same sequence multiple times?

Response: The up-sampling operation will generate small permutations to the original secondary structure sequences with a controlled rate. This permutation is similar to adding salt and pepper noise described in Azzeh J. *et.al.* (Journal on Informatics Visualization, 2018, 2(4): 252-256.), but modified specially for secondary structure sequences. We apply this up-sampling operation to generate additional sequences within individual GT families until there are 2000 sequences for each family. This is now clarified in the revised version (Lines 464468).

Reviewer #3: (Expertise: Structure and function of glycosyltransferases)

Summary: The authors employ a deep learning model to predict structural folds for glycosyltransferases (GTs) from primary sequences. GTs comprise an enormous class of enzymes that have been classified into 114 CAZy families based on sequence similarity. To date three major structural folds have been identified for glycosyltransferases; GT-A, GT-B and GT-C as well as a lysozyme type fold for family 51 peptidoglycan polymerases. The authors have previously identified a common scaffold for GT-A fold enzymes using sequence mining methods, however, these methods are not applicable to the other fold types (Taujale et al, 2020). In the present study, they achieve an accuracy of 96% based on fold prediction and 77% on family classification in a representative set of sequences. The GT landscape showed distinct clusters for the respective GT folds. Two substructures in the GT-A fold and three for GT-B and GT-C folds were identified in the clusters. The structural basis for the separations in the two GT-A fold substructures are attributed differences in helices in a hypervariable region and at the C-terminus. Patterns for the three GT-B and GT-C substructures were also determined including conserved cores. Thirty CAZy families with unknown folds were predicted to adopt known folds or their variants including five that were predicted to have novel folds. Two of these have been structurally confirmed while the three others are members of CAZy family 91, 96 and 97.

I am enthusiastic about the computational methods developed and their applications to mine sequence data. Patterns were identified for conserved GT fold clusters and subtypes within these clusters and novel fold types were predicted. To my knowledge global analyses are unprecedented for the GT-B and GT-C folds. My enthusiasm is tempered by a lack of insight into GT evolution or function although pattern identification is a step in determining evolutionary relationships

Response: We thank the reviewer for acknowledging the importance of our work to the glycobiology community. In the revised version, we emphasize the novelty of uncovering novel folds using secondary structure representations, and the common substrate binding regions identified between divergent GT-A fold families using CAM mappings. Our classification of the GT-B and GT-C serves as a conceptual starting point for generating new hypotheses for functional studies and inferences regarding GT evolution will come from

further testing of specificity within representative members of the classification framework in the future. Likewise, our prediction of new GT folds will enable new structural studies to expand the repertoire of GT families. These points are now emphasized in the revised version.

Reviewers' Comments:

Reviewer #1:

Remarks to the Author:

I'm satisfied

Reviewer #2:

Remarks to the Author:

The authors have addressed most of my concerns. One new concern:

One concern is the UMAP plot in Supplementary Figure 2. Given UMAP can give radically different results depending on parameters (`n_neighbors`, `min_dist`), it's a little concerning using UMAP visualization to argue that one method learns better embedding compared to the other. Page 25 - Line 560, the authors write "UMAP was performed with multiple sets of parameters to find families 561 that most consistently grouped together." Was the same scan of params performed for all the methods? A potential concern of readers might be that one tried really hard to get a good visual for their method, but not so hard for the other methods. I would highly recommend including a scan of params (with multiple replicates) of umap visualizes for all the methods as SI figures.

Minor comments:

page 22- line 490: "GBDT" acronym this does not appear to be defined in the main text.

REVIEWERS' COMMENTS

Reviewer #1 (Remarks to the Author):

I'm satisfied

Response:

We thank the reviewer for all the valuable feedback and comments throughout the review process.

Reviewer #2 (Remarks to the Author):

The authors have addressed most of my concerns. One new concern:

Comment #1:

One concern is the UMAP plot in Supplementary Figure 2. Given UMAP can give radically different results depending on parameters (n_neighbors, min_dist), it's a little concerning using UMAP visualization to argue that one method learns better embedding compared to the other. Page 25 - Line 560, the authors write "UMAP was performed with multiple sets of parameters to find families

561 that most consistently grouped together." Was the same scan of params performed for all the methods? A potential concern of readers might be that one tried really hard to get a good visual for their method, but not so hard for the other methods. I would highly recommend including a scan of params (with multiple replicates) of umap visualizes for all the methods as SI figures.

Response:

We thank the reviewer for providing valuable comments and feedback to improve the manuscript. We did perform the same scan of parameters on all the methods and acknowledge the importance of clarifying this for the readers. We have included the projections for these scans (including replicates) in the revised Supplementary Fig. 2 and Supplementary Fig. 3. These grid search results also demonstrate the reproducibility of fold level clustering based on the CNN-attention embeddings. However, in the transformer esm-1b based embeddings, sequences from the same fold type are scattered and do not always form distinct clusters.

Minor comments:

Comment #2:

page 22- line 490: "GBDT" acronym this does not appear to be defined in the main text.

Response:

We have added the definition for the GBDT acronym (gradient boosting decision trees) in the main text.